# Microbiota-Derived Metabolites in Tumor Progression and Metastasis

**DOI:** 10.3390/ijms21165786

**Published:** 2020-08-12

**Authors:** Tania Rossi, Daniele Vergara, Francesca Fanini, Michele Maffia, Sara Bravaccini, Francesca Pirini

**Affiliations:** 1Biosciences Laboratory, Istituto Scientifico Romagnolo per lo Studio e la Cura dei Tumori (IRST) IRCCS, 47014 Meldola, Italy; tania.rossi@irst.emr.it (T.R.); francesca.fanini@irst.emr.it (F.F.); sara.bravaccini@irst.emr.it (S.B.); 2Department of Biological and Environmental Sciences and Technologies, University of Salento, 73100 Lecce, Italy; daniele.vergara@unisalento.it (D.V.); michele.maffia@unisalento.it (M.M.)

**Keywords:** microbiota-derived metabolites, metastasis, tumor progression, tumor microenvironment

## Abstract

Microbial communities and human cells, through a dynamic crosstalk, maintain a mutualistic relationship that contributes to the maintenance of cellular metabolism and of the immune and neuronal systems. This dialogue normally occurs through the production and regulation of hormonal intermediates, metabolites, secondary metabolites, proteins, and toxins. When the balance between host and microbiota is compromised, the dynamics of this relationship change, creating favorable conditions for the development of diseases, including cancers. Microbiome metabolites can be important modulators of the tumor microenvironment contributing to regulate inflammation, proliferation, and cell death, in either a positive or negative way. Recent studies also highlight the involvement of microbiota metabolites in inducing epithelial–mesenchymal transition, thus favoring the setup of the metastatic niche. An investigation of microbe-derived metabolites in “liquid” human samples, such as plasma, serum, and urine, provide further information to clarify the relationship between host and microbiota.

## 1. Introduction

Recently, the analysis of large annotated cancer datasets led to the definition of a unique microbial signature in tissue and blood in different cancer types [1], and the number of researches that highlight the influence of microbiota in human malignancies is exponentially increasing [2,3]. However, the contribution of microbiota to cancer progression remains unclear, in particular for its dual role of promoting or inhibiting cancer progression. Indeed, microbes at the level of aerodigestive malignancies can influence tumor growth and simultaneously exert anti-inflammatory activity, suggesting a huge and complex network. Garret summarized this complexity, defining three major ways of microbiota contribution to carcinogenesis: (i) modulating the balance between cell proliferation and death, (ii) steering the immune system, and (iii) influencing the metabolism of the host [4].

The microbiota exerts its role communicating with its host through hormonal intermediates, metabolites, secondary metabolites, proteins but also toxins, which in turn are affected by diet, lifestyle, and disease. These small molecules can be de novo synthesized or metabolized from both exogenous or endogenous compounds and can enter the systemic circulation and influence distant organs. Moreover, the presence of cancer has been associated with intestinal microbiota dysbiosis, which seems to affect the production of metabolites and the host–microbiota communication [5,6]. For instance, enterotoxigenic *Bacteroides fragilis*, frequently found stool of patients when compared to controls [7], is supposed to be involved in colorectal cancerogenesis through the production of a toxin associated with an inflammatory state [8].

Trying to understand this relationship has shifted the interest of microbiota study from a phylogenetic approach to the identification of functionally active species that have an impact on the host. This is also because the composition of the microbiota varies greatly from person to person, even in healthy individuals, and does not necessarily provide functional information [9,10]. Accordingly, a functional metabolic approach based on the study of bacteria-derived metabolites plays a critical role because the metabolites can influence the homeostasis of the gut as well as areas far from it [11,12], as demonstrated by high-resolution mass spectrometry studies [13].

Intestinal microbiota metabolites have been evaluated not only for their effect on the onset and development of different tumor types, but also for their value as possible biomarkers [14,15].

As a consequence, the investigation of small molecules, such as fatty acids, bile acids, and amino acids in liquid biopsy through metabolomics and proteomic analyses, might be helpful in defining the functional relationship between microbiota and host and how it can influence the progression of some pathological conditions such as cancer. In this review, we present an overview of the role of bacteria metabolites or molecules that can derive from host metabolism and have shown ability to modulate carcinogenesis, focusing on those which could be involved in progression, invasion, and considered as potential biomarkers.

## 2. Microbiome Contribute to Cancer Hallmarks Modulating Tumor Microenvironment (TME)

Carcinogenesis is a complex multistep process in which the transformation of normal somatic cells into cancer cells is principally due to the accumulation of mutations resulting in genetic instability, enhanced proliferation, inhibition of apoptosis, tumor promoting inflammation, invasion, and metastasis, induction of angiogenesis, immune system escape, and activation of epithelial mesenchymal transition (EMT) [16,17]. However, the role of microenvironment in supporting tumorigenesis has recently acquired fundamental interest. The TME is composed by several cell populations, in particular fibroblasts, immune cells (T-, B-, and NK-cells, myeloid cells and others), and vasculature-associated cells (endothelial cells) as well as adipocytes, pericytes, and others recruited by the malignant tumor cells through the secretion of cytokines, stimulatory growth factors, and chemokine [18].

TME cells are reprogrammed by the tumor and in turn secrete growth-promoting signals, supply the tumor cells with nutrients and prepare the surrounding environment for proliferation, local invasion and metastasis [19].

The communication between tumor cells and TME is mutual, highly dynamic and exert a strong influence on all the cancerogenesis steps, but TME and the tumor also interact systemically with the entire organism, including the microbiota.

The microbiota, through its metabolites, can shape the TME [2,3] and influence the TME-tumor crosstalk by acting on the component of the TME and modulating all the most relevant tumor promoting function: inflammation, angiogenesis, metabolism and EMT.

For example, deoxycholic acid (DCA) is able to enhance the transcription of enzymes in cancer associated fibroblasts (CAFs) [20]. Lithocholic acid (LCA) can control T-Helper 17 (Th17) and regulatory T cells (Treg) differentiation [21], while lipopolysaccharide (LPS) can act on epithelial cells, promoting the induction of the EMT [22,23], and is also able to activate the vascular endothelial growth factor (VEGF) receptor, inducing angiogenesis [24].

On the basis of these evidences, microbiota acquires a fundamental role, even at a distance, in the fate of the tumor. Again, most of the metabolites are able to activate different cell receptors or act differently depending on the concentration and context, thus resulting both as friends and foes.

## 3. Toxin: Lipopolysaccharide (LPS)

LPS is a glycolipid, part of the outer membrane of the envelope of Gram-negative bacteria, whose role is to protect them from environmental threats, such as antibiotics and bile salts.

Among bacterial metabolites it is the one that have the most negative effect on carcinogenesis and metastasis. Due to its location in bacterial membrane, LPS is responsible of microbial pathogenicity as it interacts with the immune system cell populations [25]. LPS secreted by bacteria is able to trigger the host immune response through a cascade of LPS receptors, such as LPS binding protein (LPSBP), cluster of differentiation (CD)14, and Toll-like receptor 4 (TLR4). Finally, the activation of the transcription factor nuclear factor kappa-light-chain-enhancer of activated B cells (NF-kB) and other cytokines among which tumor necrosis factor-α (TNF-α), interleukin (IL)-1b, IL-6 and IL-12 is allowed, thus creating an inflammatory environment [26]. Dysbiosis of Gram-negative microbes drive higher LPS levels in serum, a condition commonly found in diabetes patients as an inflammatory biomarker [27]. Moreover, an increase of Gram-negative microbial subgroups was observed also in colorectal cancer (CRC), suggesting the involvement of their products, including LPS, in carcinogenesis [28].

LPS has been shown to have a promoter role in the induction of EMT as well in invasion and metastasis in human intrahepatic biliary epithelial cells by upregulating the expression of transforming growth factor beta 1 (TGF-β1) as a result of biliary infection. However, this process remains reversible because EMT can be blocked through TGF-β1 inhibition, but also by using statins, driving the downregulation of TLR4 expression and NF-kB phosphorylation [22,23] (Figure 1).

Recently, high LPS secretion had been associated with Cathepsin K (CTSK) overexpression and secretion in colorectal cancer. CTSK is a lysosomal cysteine protease which participates in numerous physiological processes and has been also identified as a mediator of imbalance of gut microbiota and CRC metastasis. In fact, it can stimulate the migration and motility of CRC cells binding TLR4 to enhance the M2 macrophages polarization and the secretion of cytokines by M2 tumor-associated macrophages [32].

Among the promoted transcripts of the cascade of receptors triggered by LPS, VEGF is well known to increase the angiogenesis and the metastatic behavior in different cancer types. For instance, a study showed that in pancreatic cancer (PC) TLR4 and VEGF were upregulated and their expression was positively correlated with microvessel density and with neo-angiogenic activity through the activation of the PI3K/AKT pathway [24] (Figure 2).

Similarly, LPS stimulation of breast cancer (BC) cell lines MCF-7 and MDA-MB-231 induce metastasis by activating TLR4. In addition, metastatic-related genes are subject to transcription as a result of β-catenin signaling activity through the AKT/PI3K pathway activation [35]. It has been reported an increase of LPS in CRC tissues with respect to the normal counterpart, even in the presence of lymph node metastasis. Cell motility, lymph angiogenesis and lymph node metastasis are promoted by LPS through the upregulation of VEGF-C and the activation of the signaling NF-kB/JNK [28]. By the way, VEGF was observed at higher levels in the sera of LPS-treated mice, deepening the involvement of LPS in the occurrence of BC lung metastasis through an inflammation-driven process [36].

## 4. Secondary Metabolites

### 4.1. Secondary Bile Acids (sBAs)

sBAs derive from the bile acids, which are product in the liver from cholesterol and then undergo to gut microbiota metabolism to produce their unconjugated forms, thanks to several bacterial genera, such as *Bacteroides*, *Clostridium*, and *Lactobacillus*. sBAs exert their function as ligands by interacting with several receptors, like farnesoid X-receptor (FXR), muscarinic receptor, Takeda G-protein-coupled receptor 5 (TGR5), and G-protein-coupled receptor (GPCR), thus regulating different process.

### 4.2. Deoxycholic Acid (DCA)

DCA is a metabolite derived from the metabolism of unabsorbed primary bile acids by gut microbiota. The involvement of metabolites such as DCA in CRC carcinogenesis is known since the early 2000s, as it has been demonstrated to be implicated in ERK and PKC signaling stimulation, as well as in the disruption of the tumor-suppressor p53 [37]. Indeed, studies refer the ability of DCA to act as transcriptional activator in cancer associated fibroblasts (CAFs) of the enzyme cyclooxygenase COX-2 [20], so having an influence on TME by increasing the invasiveness and proliferation of cancerous cells. In particular, the activation of COX-2 is associated with a higher production of prostaglandins, such as prostaglandin E2 (PGE2) which is a key mediator in fibrotic processes, typically produced during inflammation [38] and in some malignancies such as CRC, ovarian and pancreatic cancer. DCA is also involved in the constitutive activation of the pro-tumorigenic epithelial growth factor receptor (EGFR) mitogen-activated protein kinase (MAPK) pathway in HT-29 CRC cell line, which takes place through calcium signaling [39]. However, DCA was shown to be present at higher levels in plasma of breast cyst fluid of postmenopausal patients affected by BC [40], meaning that DCA can be involved in the carcinogenesis of organs reached through the bloodstream. Furthermore, it has been proven that high levels of microbial derived DCA contributes to liver carcinogenesis by inducing the so-called senescence-associated secretory phenotype (SASP) in hepatocytes. The onset of SASP in turn allows the production of pro-inflammatory and tumorigenic compounds, which have been demonstrated to promote hepatocellular carcinoma (HCC) development in mice treated with carcinogens. Moreover, blocking DCA production and reducing gut microbial communities prevents the development of HCC, showing DCA potency as a target [41].

### 4.3. Lithocholic Acid (LCA)

LCA has been described in cancer with contrasting results. While it was proved to have tumor suppressing functions in BC, in particular by interacting with TGR5 [42], in CRC, it was described as a potent tumor promoter by activating multiple signaling pathways. LCA seems to slow down proliferation by eliciting oxidative stress through the inhibition of Nuclear factor erythroid 2-related factor 2 (NRF2) activation, thus driving to cytostatic functions with a correlation with prognosis [43]. LCA can avoid metastasis [42] and improves the tumor immune response by regulating the balance of TH17 and Treg cells [21]. Moreover, LCA has been shown to have pro-apoptotic functions in MCF-7 and MDA-MB-231 BC cell lines, suggesting a potential therapeutic role in BC [44]. Conversely, LCA stimulates angiogenesis and metastasis in HCT116 CRC cell lines through the activation of extracellular signal-regulated kinases (Erk)1/2 and simultaneous reduction of Signal transducer and activator of transcription 3 (STAT3) phosphorylation, thus enhancing as a consequence the expression of IL-8 which is known to be upregulated in the serum of CRC patients [34] (Figure 2). In SW620 CRC cell lines, LCA upregulated via Erk1/2 the expression of the urokinase-type plasminogen activator receptor (uPAR), which is associated with invasive and metastatic behavior in several cancer types [45]. However, both DCA and LCA have shown to be risk factors in CRC as they have been reported to induce the cancer stem cell (CSC) and the upregulation of stemness-related factors in normal human colonic epithelial cells (HCoEpiC) [46].

## 5. Proteins: Polyamines (PAs)

PAs are small polycationic molecules and are involved in multiple cellular processes. The most known in human are spermine, spermidine, cadaverine, and putrescine, which are produced from ingested foods and microbiota metabolism at the level of the lower part of the intestine, then delivered in the bloodstream [47]. Overall, PAs are involved in a plethora of biological processes, among which gene expression regulation, cell proliferation and cellular stress, so that an altered concentration of them can drive to pathological conditions [48]. An increased production of polyamine was already described in proliferating cells, in particular cancer cells, as well as in the urine and blood of cancer patients [49]. However, recently, an integrated approach combining 16s rRNA sequencing and metabolomics reported different metabolites in feces of CRC patients and healthy volunteers as well as a less diversity of microbial species in the CRC cohort. Moreover, polyamines such as cadaverine and putrescine showed a higher abundance in the CRC group with respect to the healthy cohort, suggesting a role as potential biomarkers [50]. Along with these aspects, studies on pancreatic ductal adenocarcinoma (PDAC) in mouse models have showed that the early dysbiosis occurrence is responsible for the high concentration of polyamines in PDAC mice and patients, suggesting the possibility to measure these metabolites for early detection purposes [51].

### Cadaverine (CAD) and Putrescine

Among the PAs, CAD seems to have a benign role in human physiology, globally. Particularly, CAD is originated through the decarboxylation of lysine by the lysine decarboxylase (LDC) enzymes, often expressed by microbes of the gut [52], where it was shown to protect the mucosa against enterotoxins [47,53]. Furthermore, CAD has been described to also act as a tumor-suppressor in BC as its biosynthesis was proven to be downregulated in the first stages, associated with a decrease of CAD-producing microbes [29,42]. In BC, the tumor suppressor role of CAD was demonstrated in vitro by Kovacs and colleagues [29]. In fact, CADs are able to inhibit EMT, cellular movement chemotaxis, and metastasis (Figure 1), and confirming their ability to orchestrate breast carcinogenesis, higher levels of LDC enzymes were identified in patients with higher survival [29]. Another enzyme, ornithine decarboxylase (ODC), which is responsible for the production of putrescine, has been resulted upregulated in many cancers among which BC, in which it is associated with progression, metastasis, and estrogen receptor α (ERα) expression [54], suggesting a possible role for putrescine as a tumor-promoter PA.

## 6. Fermentation Products and Catabolites

### 6.1. Short Chain Fatty Acids (SCFAs)

One of the most common products of metabolism are short-chain fatty acids (SCFAs), which derive from the saccharolytic fermentation of non-digestible carbohydrates. SCFAs are mainly constituted by acetate, propionate, and butyrate, and are tightly involved in the regulation of host metabolism, immune system, cell proliferation, cell invasion and apoptosis, principally with a positive effect. SCFAs are able to shape intestinal microbiota, influencing its physiology first by protecting it and exerting an anti-inflammatory function [55,56] with consequences for intratumoral inflammation.

SCFAs are ligands of G protein coupled receptors (GPCRs), in particular, propionate and butyrate are the primary agonist of free fatty acid receptors 2 and 3 (FFAR2 and 3) also known as GPR43 and GPR41 respectively [57,58]. Recent works suggested that FFAR2 and FFAR3 played a role in tumor suppression [30,59]. In fact, the activation of FFAR2 by propionate in mice transplanted with Bcr-Abl-transfected BaF3 cells reduced the systemic inflammation in vivo, but also the proliferation rate of BaF3 in vitro [60]. Another study in a human colon cancer line model, reported an increase of apoptosis after SCFA exposure and restoration of FFAR2 receptor [61]. Some in vitro studies also highlight the protective potential of SCFAs in the formation of metastases. Binding to FFAR2 receptor, SCFAs inhibit the Hippo-yap pathway increasing the expression of adhesion protein E-cadherin, and inhibit the MAPK signaling by binding the FFAR3 receptor. The final effect is the reduction of the invasive potential of BC cells by inducing a mesenchymal–epithelial transition and regulating proliferative pathway, limiting metastasis formation [30] (Figure 1).

SCFAs also inhibit the histone deacetylases (HDACs), exerting an epigenetic function. In this view, SCFAs can suppress the EMT and prevent the expression of DNA repair protein compromising the integrity of the DNA [62]. Interestingly, SCFAs seemed to be associated with the response to immunotherapy. In a study that analyzed the presence of SCFAs in feces and plasma samples of patients with solid cancer tumor, the concentration of these molecules has been showed correlated with clinical response to immunotherapy by a dose dependent association with Programmed death-ligand 1 (PD-L1) efficacy [63].

Moreover, HDACs inhibition enhanced the expression of PD-L1 and PD-L2 in vivo and in vitro in melanoma cells, increasing the efficacy of immunotherapy [64].

#### 6.1.1. Butyrate and Sodium Butyrate (NaB)

Butyrate is one of the most characterized SCFAs. It is a product of anaerobic bacterial fermentation of non-digestible carbohydrates and it is able to maintain an anti-inflammatory condition in the colon, spreading a tumor suppressor effect by regulating CD4+ [65] and CD8+ [66] Treg cells or by interacting with the GPR41 and GPR43 receptors [67]. Butyrate can also down-regulate or up-regulate the expression of genes involved in carcinogenesis and progression such as *WNT* [68,69,70] and Endocan [71], inhibiting cell proliferation and tumor progression. Several studies also identify in NaB a therapeutic potential. Indeed, the administration of NaB to mice with CRC liver metastasis, repressed metastasis by improving host immune response and modulating the gut microbiota [71]. Moreover, in colon cancer cell line HT29 NaB were demonstrated to modulate the expression of the angiogenesis promoters VEGF and hypoxia-inducible factor (HIF)-1α [33] (Figure 2).

On the contrary, in a mouse model carrying mutation of Adenomatous Polyposis Coli (*APC*) and MutS homolog 2 (*MSH2*) genes, butyrate showed the capability to promote cell proliferation by stimulating the hyperproliferation of deficient *MSH2* epithelial cells and disregulating the Wnt–β-catenin activity [72]. This finding suggested that the effect of SCFAs could depend also by the host genotype emphasizing the need to deepen this aspect.

#### 6.1.2. Acetate

Acetate is a two-carbon monocarboxylic acid and is the most produced SCFA reaching relatively high concentration in the blood of mammalians, ranging from 50–200 μM or more [73]. Acetogens bacteria, such as *Blautia hydrogenotrophica*, are able to produce acetate from pyruvate and also through the Wood–Ljungdahl pathway [12]. Acetate is one of the major sources of energy during hypoxia and other abnormal conditions such as cancer. Unlike other SCFAs, acetate is not a HDACs ligand, but under stress conditions, it is used to produce acetyl-CoA which plays an important role in histone acetylation and gene expression regulation [74] acting as an epigenetic regulator [75]. Even acetate can play a double role in cancer progression and metastasis. Indeed, by the binding with GPR43, it can modulate Treg cells and induce anti-inflammatory effects [76,77]. Instead, under metabolic stress, acetate contributes to cancer proliferation and metastasis [78]. A possible mechanism of action of acetate has recently been discovered. In fact, acetate is able to increase Snail Family Transcriptional Repressor 1 (SNAI1), a zinc finger protein involved in downregulation of the expression of E-cadherin and mediator of the EMT, and Acyl-CoA Synthetase Short Chain Family Member 2 (ACSS2) under glucose limitation in renal carcinoma cells [31] (Figure 1).

### 6.2. Microbial Tryptophan Catabolites (MTC)

Tryptophan is one of the essential amino acids introduced through the diet and metabolized at the level of the small intestine, but it partially reaches the colon where it can be catabolized by the microbiota. It is well known that some species of bacteria, both Gram-negative and Gram-positive, are able to catabolize tryptophan converting it in indole and other derivatives, like 3-methylindole (Skatole), indole ethanol (IE), indolelactic acid (ILA), indoleacetic acid (IAA), indoleacrylic acid (IA), indolepropionic acid (IPA), and tryptamine.

However, studies that correlate the composition of the gut microbiota with the concentration of these catabolites are still needed. Although it is not clear which is the most abundant microbial tryptophan catabolite, they have been already identified in urine, faecal samples, serum, and Indole seemed to be the most represented, together with IAA and IPA [79], and they have been suggested as important signal molecules not only for bacteria physiology, but also for the host and microbiota interaction.

In fact, MTCs are antimicrobial agents, they can affect the mucosal homeostasis, influence insulin secretion, and affect the immune system through activation of cellular receptors. In particular, MTCs are known to be Aryl Hydrocarbon Receptor (AhR) ligands [80,81,82]. When the receptor binds to ligands, the AhR translocates to the nucleus promoting the transcriptions of various targets genes.

The AhR receptor is expressed by the immune system cells [83] but also by several human tumors such as breast, liver, lung, gastric, pancreatic, prostate, urothelial, ovarian cancers, T-cell leukemia, glioma, and medulloblastoma [84,85]. Recently, it has been reported that its activation altered innate and adaptive immune responses [86,87,88], but there were also evidence of the involvement of AhR in cancer initiation and metastasis [84,89,90,91,92]. A study conducted in thyroid carcinoma tissue and cell lines showed an increase in AhR expression which led to the upregulation of SLUG, which in turn repressed E-cadherin expression, an epithelial marker, and upregulated N-cadherin and fibronectin (mesenchymal markers). Other authors reported the involvement of AhR in neuroblastoma and inflammatory BC progression by promoting the establishment of an immunosuppressive TME and EMT [90,93,94].

MTCs, in particular Indole, IPA and IA, can influence the intestinal epithelial barrier maintaining its structure and function also by activation of the Pregnane X Receptor (PXR), a receptor involved in the elimination of xenobiotics and endobiotics. PXR can also affect cancer growth, progression and chemoresistance by regulating the expression of genes implicated in proliferation, metastasis, apoptosis, inflammation, and oxidative stress [95]. However, its effect is dependent on the cancer tissue in which the receptor is expressed as well as cellular context [96,97], making the network even more complex. For example, in BC and endometrial cancer, PXR was described with tumor suppressor activity by inducing apoptosis [98,99]. Conversely, PXR overexpression shows to have an anti-apoptotic power in HCT116 and HepG2 cancer cell lines, suggesting a tumorigenic role of the receptor in tissues associated with high levels of metabolism, such as liver and intestine [100,101]. Regarding the intestinal tissue, PXR has been suggested as a potential target by exploiting a microbial mimicry metabolite strategy. Indeed, the treatment with functionalized indoles led to repression of inflammation in LS174T and Caco-2 cell lines, human duodenum-derived organoids and mouse models [102]. Nevertheless, due to the tissue-specificity of PXR, targeting this receptor may be a promising yet difficult mission.

On the basis of these results, it can be assumed that since the MTC are ligands of the AhR and PXR, they can also have an effect on progression and metastasis, nonetheless the correlation between microbiota composition and the type and concentration of these metabolites in cancer still need to be confirmed. Moreover, AhR activation is ligand-specific and the AhR affinities and specificity for tryptophan catabolites are different in mice and humans [82], thus questioning the validity of the mouse model for human carcinogenesis.

## 7. Conclusions

Microbiota-derived metabolites have gained growing interest since they were considered as key actors in human–microorganism and microorganism–microorganism crosstalk. Their fundamental role in maintaining human physiology, cellular metabolism, and shaping the immune system has now been recognized. The mutualistic coexistence between humans and microorganisms led to the concept of holobiont, which underlies a strong influence of microbiota on host biology, ecology, and evolution, but also on health and disease. While the role of intratumoral microbiota in cancerogenesis is not yet well understood, the gut microbiota can influence the genesis and progression of cancer even from a distance. The microbiota-derived metabolites can reach the tumor site through the circulation and become an integral component of the TME. Although not much investigated, their contribution in the occurrence of metastasis is no less important. Recently, Rosean and colleagues showed that perturbation in the gut microbiome can affect tumor dissemination and promote BC metastasis in a mouse model [103].

However, much of the information concerning the existence, the influence on TME and the contribute to metastasis onset of metabolites has been derived largely from in vitro or in vivo studies, as summarized in Table 1. The employment of cell lines and germ free (GF) mice, enable one to understand the great potential of the microbiota, but these models are not representative of the great complexity of the human organism and microbiota. Nonetheless, GF mice exhibit a different immune system, gastrointestinal system, metabolic rate, and dietary habits.

To increase the list of microbiota metabolites and to better define their functional role, new bioinformatics and computational tools have been introduced [104]. The application of these system-level approaches could help to identify a large number of microbiota metabolites thus contributing to clarify how the microbiota communicates with tumor and metastases and to better define the TME-microbiota interaction. A dynamic interplay exists between microbiota and tumor. Microbiota releases metabolites that can influence tumor behavior and vice versa. This mutual communication is not yet completely explored. Indeed, signals released from the tumor can also modulate the microbiota itself, possibly inducing or contributing to dysbiosis. Supporting this hypothesis, a recent paper combined metagenomic and metabolomic data with mechanistic models to suggest that metabolic alterations in the TME are a major component in shaping the CRC microbiome [105].

Overall, we believe that a thorough investigation of human intratumoral microbiota and microbial-derived products using integrated omics approaches can also shed light on the ability of microbes residing in tumor tissue to contribute to the metastatic niche formation.

At the same time, investigating the role of metabolites can be helpful to characterize new targets for anti-cancer therapy acting on the gut–cancer axis. The bacterial redox protein azurin is able to enter human cancer melanoma cells (UISO-Mel-2) and induce apoptosis by binding P53 in vitro, but it is also able to induce tumor regression in vivo [106] and may be potentially used in cancer treatment. Antagonists of TLR4 have been proposed for HCC prevention, since its binding to LPS has been known to promote tumorigenesis in patients with chronic liver disease, through the release of several pro-inflammatory cytokines [14,107]. Otherwise, the administration of microbial metabolites with a predictive function can be tested, such as the treatment with a SCFAs mixture that drove to the reduction of incidence of colitis-derived CRC as well as the decrease of inflammation through the downregulation of the expression of the pro-inflammatory cytokines IL-6, IL-17, and TNF-α in mice [108]. Not to underestimate is also the fact that the presence of microbial-derived metabolites in the liquid compartments is less investigated in oncologic patients.

Last but not least, dietary interventions promoting healthy eating to re-establish the functionally good microbiota could potentially counteract the evolution of cancer or boost the effect of therapies.

In conclusion, a thorough knowledge is mandatory to understand the prognostic value of circulating metabolites as well as their possible predictive role, in order to lay the basis for an awareness of the importance of the microbiota as an actor involved in pathological processes, including carcinogenesis and metastasis.

## Figures and Tables

**Figure 1 ijms-21-05786-f001:**
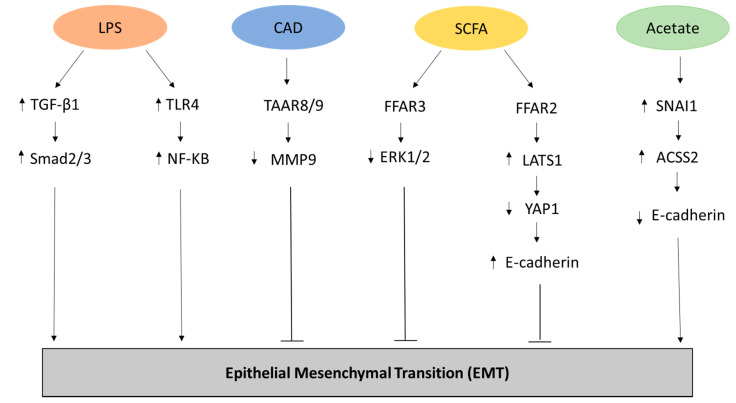
Representation of the metabolites and related pathways involved in the Epithelial-Mesenchymal transition (EMT) process in cancer. Lipopolysaccharide (LPS) has been shown to promote EMT through the upregulation of transforming growth factor beta-1 (TGFβ-1) and Mothers Against Decapentaplegic Homolog 2/3 (Smad2/3) [22] as well through the increased expression of Toll Like Receptor 4 (TLR4) and NF-KB [23] in biliary epithelial cells. Conversely, Cadaverine (CAD) can inhibit EMT in breast cancer cell lines through the activation of trace amino acid receptors 8 and 9 (TAAR8/9) modulating the expression of metalloproteinase 9 (MMP9) [29]. In breast cancer cell lines, inhibition of EMT can be carried out through the metabolic pathway of short chain fatty acids (SCFAs) that activates Free Fatty Acid Receptor 2 (FFAR2), leading to inhibition of the Hippo-Yap pathway and increased expression of adhesion protein E-cadherin, and FFAR3, resulting in mitogen-activated protein kinase (MAPK) signaling inhibition [30]. Finally, acetate can promote EMT by increasing the expression of the zinc finger protein Snail Family Transcriptional Repressor 1 (SNAI1) and Acyl-CoA Synthetase Short Chain Family Member 2 (ACSS2) in renal carcinoma cells under glucose limitation [31].

**Figure 2 ijms-21-05786-f002:**
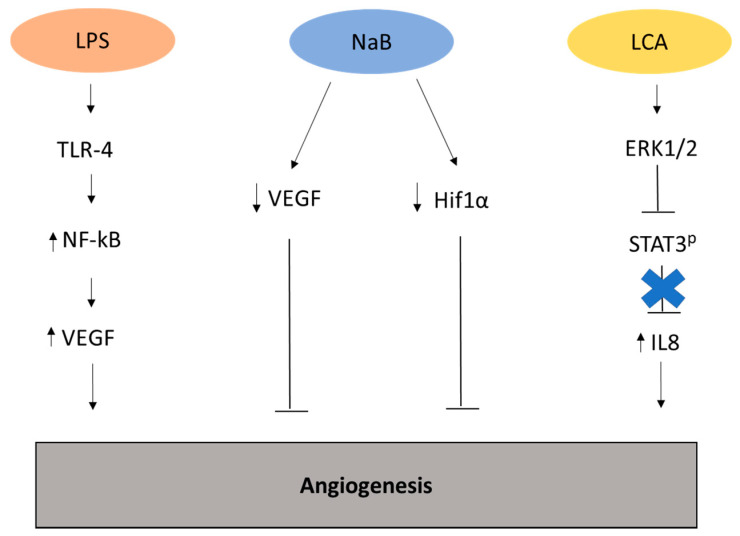
Representation of the metabolites and related pathways involved in angiogenesis in cancer. Lipopolysaccharide (LPS) can stimulate the generation of new vasculature through the upregulation of vasculature endothelial growth factor (VEGF) [28]. Conversely, the metabolite sodium butyrate (NaB) inhibits angiogenesis by modulating the expression of VEGF and hypoxia-inducible factor (HIF)-1α in colon cancer cell line HT29 [33]. Lithocolic acid (LCA) promotes angiogenesis by upregulating extracellular signal-regulated kinases (Erk)1/2, driving to the suppression of Signal transducer and activator of transcription 3 (STAT3) phosphorylation and enhanced expression of interleukin (IL)-8 in colorectal cancer cell line HCT116 [34].

**Table 1 ijms-21-05786-t001:** Metabolites involved in metastasis onset and their effects.

Type of Metabolite	Metabolite	Mechanism	Effect	Model	Organ	Reference
Toxin	LPS	TGF-β1 upregulation	EMT	H69 cells	Liver	[22,23]
		CTSK overexpression	Cell migration and motility, M2 macrophage polarization	SW480, C57 mice, CRC patients	Colon	[32]
		VEGF/VEGF-C upregulation	Microvessel density, neo-angiogenic activity, lymph node metastasis	MCF-7, MDA-MB-231, PANC-1, HUVEC, SW480, HCT116, murine models, CRC and normal tumor tissues	Breast, Pancreas, Colon, Lung	[23,24,28,32,36]
Secondary metabolites	LCA	TH17/Treg balance	Tumor immune response	mouse models	Immune system	[21]
		Erk1/2 stimulation, STAT3 phosphorylation	Angiogenesis and metastasis stimulation	HTC116	Colon	[34]
		Erk1/2 stimulation, uPAR overexpression	Invasive and metastatic behavior	SW620	Colon	[45]
	DCA	COX-2 activation	Increase invasiveness and proliferation	HT29, Caco-2, HCA7, HCT116, primary fibroblasts	Colon	[20]
Proteins	CAD	TAARs activation	EMT, cellular movement chemotaxis and metastasis inhibition	4T1-grafted mice, MDA-MB-231, SK-BR-3	Breast	[29]
	ODC	not known	Progression and metastasis	MCF-7, T47D	Breast	[54]
Fermentation products and catabolites	SCFA	Inhibition of Hippo-YAP and MAPK pathways, overexpression of E-cadherin	Reduction of invasive potential, MET	MCF7, MDA-MD-231	Breast	[30]
	Nab	VEGF165 and HIF-1α protein downregulation	Reduction of neoangiogenesis potential	HT29	Colon	[33]
		WNT upregulation	Reduction of invasive potential	HCT-116, RKO	Colon	[68,69,70]
		Endocan upregulation	Proliferation, migration and colony formation	RKO	Colon	[71]
	Butyrate	Dysregulation of the Wnt–β-catenin activity	Proliferation	Mouse models	Colon	[72]
	Acetate	SNAI1 and ACSS2 upregulation	EMT and metastasis	Human renal cell adenocarcinoma cell lines 786-O and ACHN	Kidney	[31]
	MTC	PXR binding	Inflammation repression	LS174T, Caco-2, human duodenum-derived organoids, mouse models	Colon, duodenum	[102]

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
