# Peer review of "Microbiota-Derived Metabolites in Tumor Progression and Metastasis"

_ijms, 2020, doi:10.3390/ijms21165786_

Round 1

Reviewer 1 Report

Dear authors,

I had the pleasure of reading your manuscript about “Microbiota-derived metabolites in tumour progression and metastasis”, which I found scientifically relevant and easy to follow in each section.

Although some reviews have been focused on microbiota and cancer wondering about the strict mutualism of microbes and the host as matter of physical interaction, I appreciate your will to shed more light on another interesting aspect: microbe-derived metabolites in liquid samples as potential source of tumour biomarker, clinically relevant.

While I’ve found your manuscript correctly designed, original, well written I cannot fail to underline the lack of references along the text when well-known data are expressed. Surely some concepts are now accepted by the scientific community and absolutely absorbed as facts but nonetheless needed to be referenced.

Here the lines and place where I found some missing references:

  • Lines: 45, 52, 58, 60, 205, 209, 223
  • Table 1.

Additionally, I strongly invite the authors to add acetate in Table 1 not diluting its importance among SCFAs. Indeed, it has garnered increased attention in the context of cancer as both an epigenetic regulator of posttranslational protein modification, and as a carbon source for cancer cell biomass accumulation.

Furthermore, I invite you to implement- where seems appropriate- the role of microbiota and its metabolites/toxins in skin cancer, which has never been mentioned in the manuscript. The skin represents the largest organ of our body and, together the intestine, the largest surface of interaction with microbes - thus it would be a mistake not to mention it in the context of cancer.
In fact, skin cancer is intimately influenced by the presence of skin microbiome as well as microbes are critical for tumor progression in cutaneous T cell lymphomas, for example.

I hope you will find my critics constructive for improving your manuscript.

Best regards.

Author Response

Answers of the authors to the comments of Reviewer 1

Dear Reviewer 1,

Thank you for reviewing our manuscript, we appreciate your comment and edited our manuscript as requested. Please find in below our point-by-point answers to your comments.

Comments:

1.Here the lines and place where I found some missing references:

  • Lines: 45, 52, 58, 60, 205, 209, 223
  • Table 1.

We would like to thank the reviewer for the careful and quick review of our article and for the useful suggestions that helped us to improve our manuscript. As suggested, we added references in the text, although it was not clear what we should add to Table 1. We would be grateful if the reviewer could clarify this point.

2.Additionally, I strongly invite the authors to add acetate in Table 1 not diluting its importance among SCFAs. Indeed, it has garnered increased attention in the context of cancer as both an epigenetic regulator of posttranslational protein modification, and as a carbon source for cancer cell biomass accumulation.

We thank the reviewer for the suggestions. As suggested, we added Acetate to Table 1. We also elucidated its role as a microbiota-derived metabolite in cancer in section 6.1.2 (page 9).

3.Furthermore, I invite you to implement- where seems appropriate- the role of microbiota and its metabolites/toxins in skin cancer, which has never been mentioned in the manuscript. The skin represents the largest organ of our body and, together the intestine, the largest surface of interaction with microbes - thus it would be a mistake not to mention it in the context of cancer.

In fact, skin cancer is intimately influenced by the presence of skin microbiome as well as microbes are critical for tumor progression in cutaneous T cell lymphomas, for example.

We would like to thank the reviewer for the suggestion. We implemented the text with references concerning skin cancer, as suggested, at lines 359, 361, 508.

Reviewer 2 Report

The manuscript "Microbiota-derived metabolites in tumour progression and metastasis” by the authors Tania Rossi et. al is a relevant review within the scope of the journal. The review adds new perspective to the field of cancer metabolism but it needs some major revision before it can be proceeded for publication. Suggestions about the present study are listed below.

Major Comments

  1. The manuscript should go through an extensive English revision. The grammar and choice of words are hindering the clarity of the sentences.
  2. There is a lack of central theme in this review. The review focusses on the pathways and diligently describes the metabolism of the microbial metabolites but fails to convey how these metabolites affect cancer. The author never described even in brief the processes relevant to tumor progression and metastasis which is the prime focus of the review. There is no description of different cells present in a tumor and how these metabolites affect tumor microenvironment. The authors expect its readers to be experts in tumor biology.
  3. The metabolites and pathways of metabolism are difficult to understand. Without any diagrammatic representation or simple flow chart the paper becomes too difficult to comprehend. The authors have a lot of scope to simplify represent these metabolic pathway and how they affect different processes relevant to cancer such as angiogenesis, EMT etc.

Minor comments

  1. The Introduction is divided into too many small paragraphs.
  2. Please rewrite the described metabolites into categories such as secondary metabolite, protein or toxins or other relevant category can be i) modulating the balance between cell proliferation and death, ii) steering the immune system ii) influencing the metabolism of the host as mentioned in the introduction of the review. For example LPS influences the immune system.
  3. In the heading “Microbiome metabolites that contribute to cancerogenesis and metastasis: friends and foes?”, it should be friends or foes? Instead of friends and foes?. It needs clarity.
  4. A simple diagrammatic representation can be made for key metabolites and how they affect the different steps of carcinogenesis such as angiogenesis, EMT etc.
  5. Break the Secondary bile acids section into paragraphs.
  6. In Line 123- sBAs exert their function as ligands by interacting with several receptors, thus regulating different processes. Atleast give few example of those “several receptors”.
  7. Line 133-“More in deep, the activation of this enzyme is associated with a higher production of prostaglandins such as prostaglandin E2 (PGE2) which is typically produced during inflammation and is a key mediator in fibrotic processes and in some malignancies such as CRC, ovarian and pancreatic cancer” needs clarity.
  8. Break the Polyamines section into paragraphs.
  9. Too many paragraphs in SCFA.
  10. There should be a extra column of type of metabolite in Table 1.
  11. Why are there three different columns for “Nab” in Table 1?

Author Response

Answers of the authors to the comments of Reviewer 2

Dear Reviewer 2,

Thank you for reviewing our manuscript, we appreciate your comment and edited our manuscript as requested. Please find below our point-by-point answers to your comments.

Major Comments

1.The manuscript should go through an extensive English revision. The grammar and choice of words are hindering the clarity of the sentences.

We thank the reviewer 2 for the comment, the manuscript has been revised by our editing service for the English revision.

2.There is a lack of central theme in this review. The review focuses on the pathways and diligently describes the metabolism of the microbial metabolites but fails to convey how these metabolites affect cancer. The author never described even in brief the processes relevant to tumor progression and metastasis which is the prime focus of the review. There is no description of different cells present in a tumor and how these metabolites affect tumor microenvironment. The authors expect its readers to be experts in tumor biology.

We thank the reviewer 2 for the comment. In order to make the review more understandable even for researchers in fields other than cancer biology, we added paragraph 2 entitled “Microbiome contribute to cancer hallmarks modulating TME”. Herein, as suggested, we describe in brief the hallmarks of cancer and the contribute of tumor microenvironment (TME) to tumorigenesis, the cellular types which compose the TME and how the microbiome can influence the TME.

3.The metabolites and pathways of metabolism are difficult to understand. Without any diagrammatic representation or simple flow chart the paper becomes too difficult to comprehend. The authors have a lot of scope to simplify represent these metabolic pathway and how they affect different processes relevant to cancer such as angiogenesis, EMT etc.

We thank the reviewer 2 for the comment, we added two diagrams about the contribute of some metabolite discussed to EMT, and Angiogenesis (Figures 1 and 2).

Minor comments

1.The Introduction is divided into too many small paragraphs.

We edited the introduction as requested.

2.Please rewrite the described metabolites into categories such as secondary metabolite, protein or toxins or other relevant category can be i) modulating the balance between cell proliferation and death, ii) steering the immune system ii) influencing the metabolism of the host as mentioned in the introduction of the review. For example LPS influences the immune system.

We thank the reviewer 2 for the suggestion, we divided the metabolites into toxins, proteins etc.

3.In the heading “Microbiome metabolites that contribute to cancerogenesis and metastasis: friends and foes?”, it should be friends or foes? Instead of friends and foes?. It needs clarity.

We thank the reviewer 2 for the comment. We merged that section with another (section 2) and we changed the title in “Microbiome contribute to cancer hallmarks modulating TME”. However, we still talk about “friends and foes”. We chose to use “and” because metabolites behave as friends in some condition and foes in other. 

4.A simple diagrammatic representation can be made for key metabolites and how they affect the different steps of carcinogenesis such as angiogenesis, EMT etc.

As requested by the reviewer and we added to the text two diagrams representing the contribute of some metabolites to Epithelial-to-mesenchymal transition (EMT) and Angiogenesis (Figures 1 and 2).

5.Break the Secondary bile acids section into paragraphs.

We broke the section of secondary bile acids as we re-write the metabolites in categories, as suggested by the reviewer in the minor comment 2.

6.In Line 123- sBAs exert their function as ligands by interacting with several receptors, thus regulating different processes. Atleast give few example of those “several receptors”.

We added some examples of receptors that are discussed in more detail below in the paragraph (lines 226,227), as requested by the reviewer.

7.Line 133-“More in deep, the activation of this enzyme is associated with a higher production of prostaglandins such as prostaglandin E2 (PGE2) which is typically produced during inflammation and is a key mediator in fibrotic processes and in some malignancies such as CRC, ovarian and pancreatic cancer” needs clarity.

We modified the mentioned sentence (line 242).

8.Break the Polyamines section into paragraphs.

We broke the section of polyamines as we re-write the metabolites in categories, as suggested by the reviewer in the minor comment 2.

9.Too many paragraphs in SCFA.

We edited the section of SCFAs as we re-write the metabolites in categories, as suggested by the reviewer in the minor comment 2.

10.There should be a extra column of type of metabolite in Table 1.

We added the extra column to Table 1 to classify each metabolite reported in the table, as suggested by the reviewer.

11.Why are there three different columns for “Nab” in Table 1?

We think that the reviewer refers to the three lines for NaB in Table 1 instead of columns. We made a mistake, and thus merged the three identical cells as for the other metabolites. 

Round 2

Reviewer 2 Report

The authors addressed all my questions well.